# Impact of Recreational Sports Activities on Metabolic Syndrome Components in Adolescents

**DOI:** 10.3390/ijerph17010143

**Published:** 2019-12-24

**Authors:** Fernanda Faria, Cheryl Howe, Ricardo Faria, Alynne Andaki, João Carlos Marins, Paulo Roberto Amorim

**Affiliations:** 1Department of Physical Education, Federal University of Viçosa, Viçosa, Minas Gerais 36570-900, Brazil; jcbouzas@ufv.br (J.C.M.); pramorim@ufv.br (P.R.A.); 2School of Applied Health Sciences and Wellness, Ohio University, Athens, OH 45701, USA; howec@ohio.edu; 3Department of Physical Education, Federal Institute of Education, Science and Technology of the Southeast of Minas Gerais, Rio Pomba, Minas Gerais 36180-000, Brazil; ricardoefi@yahoo.com.br; 4Department of Sport Science, Federal University of Triângulo Mineiro, Uberaba, Minas Gerais 38025-180, Brazil; alynne.andaki@uftm.edu.br

**Keywords:** accelerometer, adolescent, lifestyle, metabolic syndrome, obesity

## Abstract

We investigated the impact of a sports activities program on metabolic syndrome (MetS) components and pre-MetS among adolescents. Blood samples, blood pressure, weight, height, body mass index, waist circumference, body fat percentage, frequency of food consumption, daily time in moderate-to-vigorous physical activity (MVPA), and sedentary behavior (SB) of 92 male adolescents aged 14–18 years (16.07 ± 0.93) were evaluated. From this initial sample, 36 participants (39.1%) were diagnosed with pre-MetS or MetS and were invited to participate in the intervention program. Twelve individuals diagnosed with pre-MetS or MetS agreed to participate in a recreational sports activities program lasting 14 weeks. The pre- and post-sport program comparison showed a reduction in total cholesterol, low-density lipoprotein, and non-high-density lipoprotein (HDL), and an increase in HDL and MVPA time in the intervention group. Sports activities accounted for 42% of the MVPA daily recommendation, and at the end of the intervention period, only seven subjects maintained a positive diagnosis for pre-MetS or MetS. This study showed that recreational sports activities had a significant impact on the lipid profile.

## 1. Introduction

The increase in the prevalence of obesity and hypertension among children and adolescents is considered a public health problem with severe epidemiological and economic implications [1], and it has favored the emergence of the metabolic syndrome (MetS) in this population [2].

MetS is often defined as the clustering of three or more risk factors. It can include adiposity, hyperglycemia, hypertension, hypertriglyceridemia, and lower high-density lipoprotein (HDL) [3]. The MetS prevalence in Brazilian adolescents ranges from 2.6% [4] to 6.7% [5], possibly reaching 37.1% [5] among overweight or obese adolescents. It is critical to highlight that the prevalence of any disease or syndrome relies on the criteria being used to diagnose the disease or syndrome. However, there is no consensus on the criteria used for diagnosing MetS in adolescents [6].

Using the adult criteria, the significant rates of this disorder in the pediatric population indicate the need for early diagnosis to avoid future complications in general health [7,8]. Thus, a new approach to the adult metabolic profile evaluation has been suggested through the identification of pre-MetS. The pre-MetS is characterized by having two of the risk factors associated with the MetS period [9,10].

Pre-MetS and MetS are significantly and negatively associated with reduced physical activity and increased time spent in sedentary behavior (SB) [11]. Physical activity is considered an essential component of a healthy lifestyle and must be incorporated in everyday living as a primary strategy to prevent disease [12]. It is recommended that adolescents engage in 60 min daily of moderate-to-vigorous physical activity (MVPA) or more to obtain health benefits [13]. Simultaneously, time spent in SB—activities carried out in a sitting position, such as sitting in front a TV or computer or playing traditional videogames—is associated with high body fat percentage and MetS prevalence among adolescents [14], independent of the physical activity levels [15].

Considering the focus on lifestyle changes, multidisciplinary interventions (diet and physical activity) directed to adolescents have been conducted to reduce obesity and its comorbidities, including MetS [6,16,17,18]. However, we have not found in the literature whether recreational sports activities, which are easily applicable in the school context, are more motivating and have a higher adherence among young people, reflecting positively on the control of MetS risk factors. Thus, the objective of this study was to evaluate the impact of a recreational sports program on the components of pre-MetS and MetS in Brazilian adolescents.

## 2. Materials and Methods

### 2.1. Participants and Ethical Aspects

The sample consisted of students recruited from a full-time federal institution of technical education from the county of Rio Pomba, Minas Gerais, Brazil; according to the following inclusive criteria: to be a male resident high school student, living on the campus of the school; between 14 and 18 years of age. Of the approximately 500 students enrolled in the first, second, and third years of high school, 109 were considered eligible; however, 17 students did not perform the necessary assessments for the study. Thus, the MetS components were evaluated in a sample composed of 92 individuals. From this number, 36 were diagnosed with pre-MetS or MetS and were invited to participate in the activities program, with the final acceptance of 12 students. In this second phase, the exclusion criteria included the use of medications to control blood pressure, fasting glucose, or lipid metabolism, other previously diagnosed chronic diseases besides MetS, permanent or temporary physical disability, and caloric restriction diet.

The research project was submitted and approved by the Human Research Ethics Board from the Federal University of Viçosa (Off. Ref. No. 177.176/2013/CEPH), following the Resolution 466/12 of the National Health Council and following the Declaration of Helsinki. All the participants were volunteers and signed the informed assent form, while parents of the participants under the age of 18 signed the informed parental consent.

### 2.2. Procedure

The study’s first phase is related to a pre-MetS and MetS cross-sectional baseline screening, while the second phase consisted of a longitudinal quasi-experimental study with the participation of 12 students (10 with pre-MetS and 2 with MetS) in a 14 week recreational sports program.

In the first phase, all 92 students were evaluated for body composition, maturation status, eating habits, resting vitals, physical activity, SB, and MetS components, before the beginning of the program. Of the initial 92 students, the 12 participants in the intervention program also had their resting heart rate monitored to estimate the maximal heart rate. During the 13th week of the intervention, participants enrolled in the second phase had their physical activity and SB evaluated again, while the other evaluations were performed in the 14th week.

### 2.3. MetS Components

After fasting for 12–14 h, morning blood samples were collected in the healthcare sector of the institution, by qualified and trained professionals, designated by the accredited laboratory to conduct the analyzes. We collected 5 mL of blood from the cubital vein and then centrifuged it at 3400 rpm for five minutes to separate serum from other blood components. The levels of fasting glucose, total cholesterol, HDL, and triglycerides were determined through enzymatic colorimetric assay method, while insulin level was analyzed using the electrochemiluminescence method. The equipment used for the blood tests was the biochemical analyzer Awareness Technology^®^ (model ChemWell-T, Palm City, Florida, USA).

MetS and the cutoff points used for each of its components were defined according to De Ferranti et al. [19], which considers the presence of three or more of the following criteria: triglycerides concentration > 100 mg/dL; fasting glucose > 110 mg/dL; waist circumference > 75th percentile for age and sex; blood pressure > 90th percentile for age, sex, and height; and HDL cholesterol < 45 mg/dL. Pre-MetS was defined by the presence of two of these components [10].

Insulin resistance was evaluated by homeostatic model assessment (HOMA-IR), according to the equation proposed by Matthews et al. [20], and low-density lipoprotein (LDL) by the Friedewald et al. [21] equation. The non-HDL cholesterol determination occurred through the formula proposed by Srinivasan et al. [22]. The cutoff points adopted for the HOMA-IR and LDL index were those recommended by the Guidelines for the Prevention of Atherosclerosis in Childhood and Adolescence [23]; for insulin, the values established by Ten and McLaren [24] were used; and for non-HDL the recommendation of Srinivasan et al. [22] was applied.

### 2.4. Anthropometry and Body Composition

Anthropometric measures (height and weight) and skinfolds were performed according to procedures standardized by the International Society for the Advancement of Kinanthropometry [25]. All measurements were duplicated, and a third measurement was performed if the technical error of measurement was exceeded [25]. Weight was obtained to the nearest 0.05 kg using a calibrated digital scale Líder^®^ (model Leader P-200c, Araçatuba, São Paulo, Brazil) and height to the nearest 0.1 cm using a portable stadiometer Sanny^®^ (model Personal Caprice, São Paulo, Brazil). The body mass index (BMI) was calculated from the kg/m^2^ ratio, and the cutoff points proposed by the World Health Organization (WHO) were used to classify weight status [26].

Waist circumference was obtained from two distinct anatomical points, including the measurement in the region of the smallest waist circumference (SWC) [25], as well as in the midpoint between the last rib and the iliac crest (MWC) [27]. Circumference measurements were performed by the use of a flexible and inelastic metallic anthropometric tape, graduated in centimeters, and subdivided in millimeters Sanny^®^ (São Paulo, Brazil). Subsequently, the waist-to-height ratio was calculated.

Body fat percentage was estimated by the Slaughter et al. [28] equation, from the sum of triceps and medial calf skinfold (mm) using a calibrated caliper with 1 mm precision and 67 mm amplitude Lange^®^ (model TBW, Maryland, USA). The value of 20% as an indicator of body fat excess was used as recommended by Going et al. [29].

### 2.5. Maturational Evaluation

The maturational stage was obtained through the peak height velocity using the equation and the classification suggested by Mirwald et al. [30], which considers leg and trunk lengths, age, weight, and height. Peak height velocity was classified into three groups: pre-pubertal (peak height velocity ≤−1), pubertal (peak height velocity = 0), or post-pubertal (peak height velocity ≥ 1).

### 2.6. Resting Blood Pressure

Blood pressure was measured three times, after a period of 5–10 min at rest; and at least 1 min between measurements. Blood pressure was considered the average value from the last two measurements, according to the protocol proposed by the Guidelines for the Prevention of Atherosclerosis in Childhood and Adolescence [23]. The measure was performed by a certified and trained evaluator using a mercury column sphygmomanometer to the nearest 2 mmHg and a range of 10–300 mmHg (Unitec^®^, São Paulo, Brazil), a stethoscope Premium^®^ (model Rapport model, Rio de Janeiro, Brazil) and suitable cuff size for adolescents. 

### 2.7. Resting Heart Rate

Before the beginning of the intervention activities, each participant had their resting heart rate evaluated. The student was taken to the evaluation room and asked to remain at rest, in the lying position, for 10 min. The lowest heart rate recorded in this period was used as the resting heart rate. The maximal heart rate was estimated using the Tanaka et al. [31] formula.

### 2.8. Eating Habits

The participant’s eating habits were assessed using a Food Frequency Questionnaire, adapted from a regional model, with the addition of types of food provided by the institutional refectory. The items included in the questionnaire were: bread, pasta, cereal, tuber, vegetables, fruits, meat, sausage, egg, sugar, sweets, snacks, oils, fats, and seasonings. Participants were instructed to fill out the number of times per week and per day in which they consumed a particular food group. As all the participants were resident students, most of the food consumption made for them was prepared by the school cafeteria. Then, to facilitate the completion of the questionnaire, we provided a spreadsheet with the average number of times per week each food group was served. Two nutritionists reviewed the forms as they were completed for accuracy. 

### 2.9. Physical Activity and Sedentary Behavior

Physical activity and SB were assessed using a triaxial accelerometer Actigraph^®^ (GT3X model, Florida, USA). Participants received the accelerometer and were instructed to wear it at the waist, toward the midline of the thigh, on the right side of the body, near the iliac crest. The accelerometer was worn for seven consecutive days and removed only during aquatic activities (i.e., bath, swimming, and related activities) and during sleep time. The data collection was considered valid for participants who recorded at least 10 h of daily use, in at least 4 weekdays and 1 weekend day [32]. The accelerometer was initialized at 30 Hz, and its data were converted to 10 s epochs using ActiLife 5.0 software (Actigraph^®^, Florida, USA). The activity intensity classification was based on the cutoff points proposed by Puyau et al. [33], which stratifies physical activity behavior into four distinct categories: SB < 800; light activity ≤ 3199; moderate activity ≤ 8199; and vigorous activity ≥ 8200 counts/min.

Total daily energy expenditure was estimated from the proposed equation of the National Academy of Sciences Institute of Medicine Food and Nutrition Board for boys aged 3–18 years [34]. To calculate the result, we used a formula that includes age, physical activity level, weight, and height. The basal metabolic rate was estimated by the Schofield equation [35].

### 2.10. Intervention

Participants were asked to attend intervention sessions regularly. Interventions were constituted of popular recreational sports in Brazil, including volleyball, basketball, soccer, and futsal modalities, practiced 4 days a week, lasting 1 h each. Futsal modality was offered in two weekly sessions. Volleyball and basketball were provided during the same session for similar durations of two times per week. Every 15 days, the volleyball/basketball session was replaced by a five-a-side soccer game.

Participants wore heart rate monitors Garmin^®^ with the Global Positioning System (Forerunner 610 model, Kansas, USA) to determine the intensity reached during intervention activities. The average session intensity was identified using the heart rate reserve method, calculated from the difference between resting heart rate and maximal heart rate, according to the following formula: heart rate reserve % = ((AHR − resting heart rate)/heart rate reserve) * 100; in which AHR means the average heart rate during the intervention session.

The program lasted 14 weeks for a total of 56 sessions, and adolescents were asked to attend at least 70% of the total session number (29 sessions). A physical education teacher performed the intervention and was responsible for the sports practice organization, division of teams, sports rules adaptation when necessary, and activity time control. 

### 2.11. Statistical Analysis

The variables were checked for normality of distribution using Kolmogorov–Smirnov and Shapiro–Wilk tests for the first and second phases of the study, respectively. The linear regression model was applied to evaluate the association among adiposity, metabolic, hemodynamic, and anthropometric variables. Log transformations were used to normalize non-parametric data of triglycerides, HOMA-IR, and insulin. According to the data distribution, Student’s t-test or Mann–Whitney test was applied for independent samples. Moreover, in agreement with the distribution, paired Student t-test or Wilcoxon was used to evaluating possible differences in pre- and post-intervention variables. The effect size for the comparison tests was calculated using a specific formula. All statistics were performed using SPSS 20.0 (IBM Corporation, Armonk, NY, USA), and significance was set at *p* < 0.05.

## 3. Results

In the first phase of the study carried out with 92 participants, we found a pre-MetS and MetS prevalence of 34.8% (n = 32) and 4.3% (n = 4), respectively. The individuals characterized with the pre-MetS or MetS were mostly (72.2%) in post-pubertal maturation stage, 27.8% were classified as pubescent, and no individual was classified as pre-pubertal. Table 1 and Table 2 represent the components initially evaluated for diagnosing pre-MetS or MetS. Regarding the anthropometric variables, participants with pre-MetS or MetS had higher weight (*p* = 0.004), BMI (*p* = 0.021), MWC (*p* = 0.030), and SWC (*p* = 0.025). Similarly, significantly higher values were found for systolic blood pressure (*p* = 0.005), diastolic blood pressure (*p* = 0.002), insulin (*p* = 0.049), HOMA-IR (*p* = 0.000), triglycerides (*p* = 0.000), and non-HDL (*p* = 0.000) in the pre-MetS and MetS groups (Table 1). No significant differences were found in time spent on MVPA and SB between groups, while basal metabolic rate was significantly higher in subjects with pre-MetS or MetS (*p* = 0.004) (Table 2).

Table 3 depicts only the significant associations found between central adiposity measures and metabolic variables. The SWC had the best association values with HDL when compared to the other adiposity variables, explaining 11.5% of the metabolic component variation. The SWC also revealed the most significant associations with BMI, explaining 74.1% of the variation in this variable. Systolic blood pressure and diastolic blood pressure were positively and significantly associated with all measures of central adiposity.

On average participants attended 32.6 intervention sessions (77.6% of the total). Table 4 and Table 5, as well as Figure 1, show pre- and post-intervention comparisons for the 12 participants. There were no significant changes in anthropometric variables, except for height, which was statistically higher after the intervention (*p* < 0.001) (Table 4). However, the metabolic variables total cholesterol, LDL, and non-HDL were statistically lower (*p* < 0.05) after 14 weeks of the sports program. Besides that, the variables age and HDL demonstrated statistically significant increases (*p* < 0.05) when comparing the two study moments.

The intervention period significantly increased the weekly time spent in MVPA (*p* = 0.016) and the total daily energy expenditure (*p* = 0.010). Although not significant, there was a trend in increasing MVPA during weekdays (*p* = 0.07). No significant differences were found in the other variables, including SB (Table 5). 

The prevalence of pre-MetS or MetS among intervention participants decreased by 41.6%, reaching seven adolescents at the end of the program (Figure 1). The fasting glucose values were not shown since no participant showed high levels of this component during the first and second phases of the study. The intervention group reduced the MetS markers percentage for most of the evaluated parameters; except for the waist circumference measures.

Before starting the program, we calculated the maximal heart rate (197 bpm) and resting heart rate (62 bpm) average. The average heart rate during the intervention session was 134 ± 11.2 bpm. According to the heart rate values, the participants remained, on average, 25 min (≈42%) of the activity duration (60 min) within the moderate-to-vigorous intensity, with a mean of heart rate 157 ± 3.8 bpm. This time is considered an adequate amount of time in MVPA, considering the recommendation that 50% of physical education classes (approximately 30 min) should be performed at that intensity [36]. In the nutritional evaluation, no statistically significant differences were found between pre- and post-intervention.

## 4. Discussion

The definition of an internationally accepted criterion for MetS diagnosis in children and adolescents is a current debate by many researchers [37,38,39]. The diverse criteria and different cutoff points for each MetS component have provided discrepant prevalence among different countries [1,2,19]. However, regardless of the diagnostic method used, the harmful effects of pre-MetS and MetS on the cardiovascular health of children and adolescents, as well as their consequences in adult life, are well known [7,40]. 

Our study diagnosed 4.3% of the total sample (n = 4) with MetS using the adult criteria, a prevalence similar to other studies involving adolescents [37,41]. However, a higher prevalence was found for pre-MetS, in which 34.8% (n = 32) of the adolescents had at least two of the diagnostic components. This early detection, or pre-MetS classification, is essential since individuals with pre-MetS are at high risk for developing MetS, as well as cardiovascular disease and type 2 diabetes mellitus [10]. The adolescents classified with pre-MetS or MetS already presented higher values for weight, BMI, MWC, SWC, systolic blood pressure, diastolic blood pressure, triglycerides, insulin, HOMA-IR, total cholesterol, LDL, and non-HDL cholesterol, and lower HDL values (*p* < 0.05). These results illustrate the metabolic impact of this condition and also suggest negative changes in lipid and glycolytic metabolism during adolescence. However, no significant differences in fasting glucose were identified between groups (with or without pre-MetS or MetS), since this variable typically exhibits a late response compared to the other MetS components [8]. Similarly, there were no significant differences in the MVPA between the groups, a fact that reinforces the multifactorial nature of MetS, which could be related to genetic, environmental, and behavioral aspects. These results, together, highlight the need to screen groups at risk for developing chronic noncommunicable diseases that could then allow the development of intervention strategies that meet the needs of this specific population.

The literature has been cohesive in treating anthropometric variables as metabolic predictors and cardiovascular disease risk factors in Brazilian children and adolescents [8,42]. Our results corroborated with such studies and suggested a significant association among the HDL, systolic blood pressure, diastolic blood pressure, and BMI with anthropometric indicators (Table 3). Further, SWC was the best predictor of lower HDL and higher BMI values, explaining 11.5% and 74.1% of the variability in these components, respectively, while BMI was the best predictor for elevated systolic blood pressure and diastolic blood pressure, explaining 18.3% and 14%, respectively. These results reinforce the necessity and importance of anthropometric evaluation as a simple, low-cost technique that has high value in cardiovascular and metabolic risk prediction.

The findings in this study agree with other investigations [15,16,17,18,43] that present a short period of successful intervention to reduce the risk factors among adolescents. After 14 weeks of these sports intervention, the presence of risk factors for MetS decreased, resulting in a reduction of 41.7% in the diagnosis of pre-MetS or MetS. By the end of the intervention, only seven participants remained diagnosed with pre-MetS or MetS (Figure 1). In addition, the results indicated that the sports program was effective in improving the adolescents’ lipid profiles, with a significant decrease (*p* < 0.05) in several risk factors, including total cholesterol, LDL, non-HDL cholesterol; as well as increase in protective factors such as HDL, daily MVPA, and total daily energy expenditure (Table 4 and Table 5).

Studies have identified anthropometric variables as significant risk factors and health indicators, showing that modifications in these variables may reduce MetS diagnoses [43,44]. Despite the reduction in the pre-MetS or MetS prevalence and positive results in cardiometabolic biomarkers, the intervention did not result in positive changes in anthropometric measures. This fact suggests that the duration of the intervention may not have been enough to promote changes in these measures, reinforced by the literature demonstrating improvements in anthropometric measures only in interventions of at least 6 months [6,38]. Intervention studies lasting approximately 14 weeks [15,17] have reported increases in some anthropometric measures, such as BMI and waist circumference; however, the interventions included aerobic and resistance exercises, unlike our study, which used sports activities characteristic of the school environment primarily. 

The positive lipid modifications following the intervention confirm the influence of MVPA on adolescents’ metabolic health, as highlighted in the literature [45,46]. It is valid to consider that the benefits are not restricted to the minimum recommended dose of 60 min of daily MVPA since the adolescents evaluated had mean baseline values higher than 60 min daily. After the intervention, they significantly increased the regular MVPA time (Table 5), emphasizing that the higher MVPA dose leads to better health outcomes, corroborating the WHO’s statement [13].

Another critical study observation is the amount of time the adolescents spent in SB, close to 10 h daily (approximately 598 and 587 min, pre- and post-intervention, respectively) (Table 5). This high level of SB time alerted us to the harmful interactions of this behavior on body composition and metabolic variables. The literature points to deleterious effects of spending more than 2 h daily in this type of behavior in children and adolescents [14,15]. Studies involving pediatric population suggest a significant association between SB and BMI [47], waist circumference [47], triglycerides [48], and HDL [49]. However, children and adolescents who frequently interrupt their time in SB may be at lesser risk, since small breaks in SB appear to be sufficient to reduce cardiovascular risk [49]. Thus, intervention programs in adolescence, especially among those in a full-time school, should address the reduction of this behavior and create strategies to promote physical activity, even at light intensities, as an alternative to SB.

The intervention program was not intended to establish an intensity range to be maintained by adolescents. Despite that, the degree of involvement with the sports activities promoted voluntary, significant time in moderate to vigorous intensity. Approximately 42% (25 min) of the daily recommendation of MVPA was obtained during this intervention. Therefore, the physical activity ludic aspect should be contemplated in future intervention programs, especially in the school environment, to promote the adoption and maintenance of an active lifestyle by adolescents [50].

We believe that the results achieved by this study should be interpreted as the exclusive result of the physical activity promotion since no statistically significant differences were found in the frequency of food consumption. This fact was already expected due to the standardization of food provided in a boarding school environment, where most meals reported by the adolescents were those offered by the institution.

The main limitations of this study were the male participation restriction and the small number of participants in the intervention phase. However, the originality of the study stands out because, to the best of our knowledge, no intervention studies based on the promotion of recreational sports activities with Brazilian adolescents were found in the literature. As sports activities are inherent to the everyday life of schoolchildren, this increases the ecological validity of our findings. This type of group intervention should be considered as a viable, easy-to-perform, low-cost, and extremely motivational alternative for this particular population; when compared to interventions restricted to aerobic and resistance exercises program performed in closed and controlled environments. Lastly, it is suggested that future intervention studies focused on sports activities should extend the duration longer than 14 weeks and include female adolescents. We also suggested that interventions should be tailored according to the needs of this population.

## 5. Conclusions

This study suggests a high prevalence of pre-MetS or MetS among adolescents. The new pre-MetS classification was meaningful in identifying those adolescents at risk for developing MetS and should be incorporated in this population. Following the intervention, reductions in risk factors for MetS were observed, mainly in the lipid measures. The intervention reinforces that recreational sports activities contributed to increasing time spent in MVPA. Besides that, it highlight that effects are essential as preventive action and non-pharmacological therapeutic procedures for adolescents at risk of developing chronic non-communicable diseases to improve their metabolic profile. Positive lifestyle changes by means of increasing MVPA and reducing SB are the bases for MetS prevention in this population.

## Figures and Tables

**Figure 1 ijerph-17-00143-f001:**
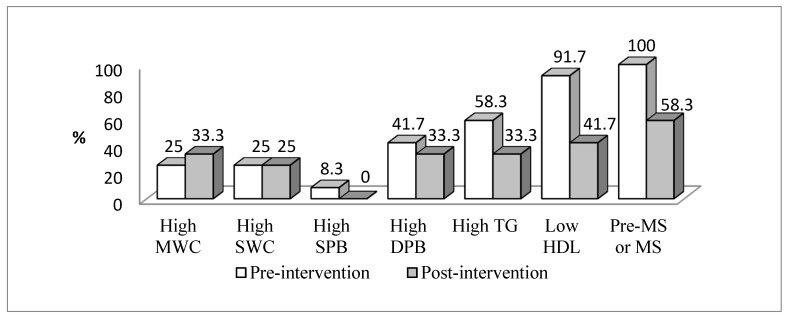
Changes in metabolic syndrome components in adolescents before and after 14 weeks of intervention.

**Table 1 ijerph-17-00143-t001:** Participants’ metabolic, anthropometric, and hemodynamic profile, with and without metabolic syndrome or pre-metabolic syndrome.

Variables	Total (n = 92)	No Pre-MetS or MetS (n = 56)	Pre-MetS or MetS (n = 36)	*p* Value	ES
Age (years) *	16.0 (14.0–18.0)	16.0 (14.0–18.0)	16.0 (14.0–18.0)	0.287	0.11
Weight (kg) *	61.3 (39.1–105)	60.6 (39.1–84.1)	65.1 (48.8–105)	0.004 ^†^	0.30
Height (m)	1.74 (±0.06)	1.73 (±0.06)	1.75 (±0.06)	0.214	0.33
BMI (kg/m^2^) *	20.5 (15.1–33.9)	20.2 (15.1–25.1)	21.2 (16.8–33.9)	0.021 ^†^	0.24
WHtR *	0.4 (0.36–0.56)	0.4 (0.36–0.5)	0.41 (0.36–0.56)	0.139	0.15
BF%*	13.4 (7.2–41.4)	13.5 (7.9–26.6)	12.5 (7.2–41.4)	0.914	0.01
MWC (cm) *	72.1 (63.0–99.4)	71.7 (63.0–86.8)	74.7 (64.5–99.4)	0.030 ^†^	0.23
SWC (cm) *	71.4 (61.4–97.1)	70.9 (61.4–84.0)	73.0 (63.6–97.1)	0.025 ^†^	0.23
SBP (mmHg)	111.6 (±10.8)	109.1 (±10.2)	115.5 (±10.7)	0.005 ^†^	0.61
DBP(mmHg)	72.5 (±7.7)	70.5 (±7.3)	75.6 (±7.3)	0.002 ^†^	0.70
HDL (mg/dL)	42.9 (±8.6)	45.8 (±8.9)	38.3 (±5.9)	0.000 ^‡^	1.00
TG (mg/dL) *	85 (36–288)	75.5 (36–110)	111 (65–288)	0.000 ^‡^	0.64
FG (mg/dL)	77.8 (±7.5)	77.1 (±7.1)	78.9 (±8.0)	0.248	0.25
Insulin (mU/L) *	4.8 (1.4–28.5)	4.4 (1.4–21.0)	6.09 (1.9–28.5)	0.049 ^†^	0.21
HOMA-IR *	0.98 (0.27–5.44)	0.85 (0.27–4.58)	1.13 (0.36–5.44)	0.045 ^†^	0.21
TC (mg/dL)	159.6 (±23.6)	155.4 (±22.8)	166.2 (±23.5)	0.031 ^†^	0.47
LDL (mg/dL)	97.6 (±19.4)	94.4 (±18.3)	102.6 (±20.4)	0.048 ^†^	0.42
Non-HDL	116.7 (±22.6)	109.5 (±19.4)	127.8 (±23.1)	0.000 ^‡^	0.86

^†^*p* < 0.05; ^‡^
*p* < 0.001. Note: BMI = body mass index; WHtR = waist-to-height ratio; BF% = body fat %; MWC = midpoint waist circumference; SWC = smallest waist circumference; SBP = systolic blood pressure; DBP = diastolic blood pressure; HDL = high-density lipoprotein; TG = triglycerides; FG = fasting glucose; HOMA-IR = Homeostatic Model Assessment; TC = total cholesterol; LDL = low density lipoprotein; Non-HDL = not HDL cholesterol; MetS = metabolic syndrome; ES = effect size. * Variables expressed as median (minimum and maximum); other variables: average (± standard deviation).

**Table 2 ijerph-17-00143-t002:** Description of physical activity variables within physical and energy expenditure.

Variables	n	Total Sample	n	No Pre-MetS or MetS	n	Pre-MetS or MetS	*p* Value	ES
MVPA (min/week)	83	70.3 (±19.8)	51	71.4 (±21.3)	32	68.6 (±17.2)	0.54	0.14
MVPA (min/weekday)	83	76.5 (±20.8)	51	77.3 (±22.0)	32	75.3 (±19.2)	0.67	0.10
MVPA (min/weekend day) *	80	48.5 (1.5–149.5)	49	51.5 (1.5–149.5)	31	48 (4.5–113)	0.61	0.05
TEE (kcal/day)	83	3095 (±582)	51	3004 (±472)	32	3240 (±708)	0.11	0.40
BMR (kcal/day) *	92	1743 (1350–2516)	56	1730 (1350–2145)	36	1810 (1521–2516)	0.004 ^†^	0.30

^†^*p* < 0.01. Note: MVPA = moderate-to-vigorous physical activity; TEE = total energy expenditure; BMR = basal metabolic rate; MetS = metabolic syndrome; ES = effect size. * Variables expressed as median (minimum and maximum); other variables: mean (± standard deviation).

**Table 3 ijerph-17-00143-t003:** Association between anthropometric indicators of obesity and metabolic variables in male adolescents (n = 92).

Variables		MWC	SWC	BF%	BMI	WHtR
HDL	β R^2^	−0.428 ^†^ 0.106	−0.487 ^†^ 0.115	−0.129 0.010	−0.860 ^†^ 0.082	53.344 ^‡^ 0.048
SBP	β R^2^	0.604 ^†^ 0.136	0.670 ^†^ 0.140	0.535 ^†^ 0.108	1.601 ^†^ 0.183	93.779 ^†^ 0.095
DBP	β R^2^	0.335 ^†^ 0.082	0.391 ^†^ 0.094	0.234 ^‡^ 0.041	0.996 ^†^ 0.140	59.009 ^†^ 0.074
BMI **	β R^2^	0.007 ^†^ 0.722	0.008 ^†^ 0.741	0.006 ^†^ 0.458	––	1.384 ^†^ 0.736

^†^*p* < 0.001. ^‡^
*p* < 0.05. ** Transformed into a log. Note: HDL = high-density lipoprotein; SBP = systolic blood pressure; DBP = diastolic blood pressure; BMI = body mass index; MWC = midpoint waist circumference; SWC = smallest waist circumference; BF% = body fat %; WHtR = waist-to-height ratio.

**Table 4 ijerph-17-00143-t004:** Anthropometric and metabolic variables at pre- and post-intervention (n = 12).

Variables	Pre-	Post-	*p* Value	ES
WHtR *	0.42 (0.39–0.56)	0.42 (0.37–0.53)	0.071	0.36
BF%	18.11 (±10.25)	18.31 (±9.72)	0.771	0.02
MWC (cm) *	74.47 (67.5–99.0)	72.57 (69.1–92.8)	0.875	0.03
SWC (cm)	75.17 (±7.55)	75.09 (±7.14)	0.896	0.01
SBP (mmHg)	114.5 (±12.19)	111.25 (±10.2)	0.082	0.29
DBP (mmHg)	75.41 (±8.37)	76.0 (±6.79)	0.737	0.07
HDL (mg/dL)	38.53 (±6.19)	52.10 (±14.25)	0.004 ^†^	1.30
TG (mg/dL) *	101.0 (65.0–286.0)	77.6 (49.0–112.5)	0.117	0.31
FG (mg/dL)	78.02 (±8.55)	83.86 (±5.92)	0.005	0.81
Insulin (mU/L) *	5.91 (1.91–25.4)	6.90 (2.58–19.7)	0.347	0.19
HOMA-IR *	1.12 (0.36–4.35)	1.35 (0.6–3.95)	0.638	0.09
TC (mg/dL)	164.01 (±27.18)	148.01 (±24.2)	0.024 ^†^	0.62
LDL (mg/dL)	103.76 (±24.44)	79.68 (±19.53)	0.002 ^†^	1.1
Non-HDL	125.48 (±26.96)	95.90 (±21.83)	0.001 ^†^	1.2

^†^*p* < 0.05. Note: BMI = body mass index; WHtR = waist-to-height ratio; BF% = body fat %; MWC = midpoint waist circumference; SWC = smallest waist circumference; SBP = systolic blood pressure; DBP = diastolic blood pressure; HDL = high-density lipoprotein; TG = triglycerides; FG = fasting glucose; HOMA-IR = Homeostatic Model Assessment; TC = total cholesterol; LDL = low density lipoprotein; Non-HDL = not HDL cholesterol; ES = effect size. * Variables expressed as median (minimum and maximum); other variables: average (± standard deviation).

**Table 5 ijerph-17-00143-t005:** Energy expenditure, physical activity, and sedentary behavior before and during the intervention period.

Variables	n	Pre-	Post-	*p* Value	ES
TEE (kcal/day)	11	2963 (2511–4423)	3494 (2886–4752)	0.010 ^†^	0.55
BMR (kcal/day) *	12	1822 (±212)	1837 (±203)	0.164	0.07
MVPA					
min/day *	11	72.5 (56.5–108.5)	81.4 (73–137)	0.016†	0.51
min/weekday	11	88 (±19.9)	100.6 (± 19.5)	0.070	0.64
min/weekend day *	10	50.4 (±27.6)	39.9 (± 29.8)	0.291	0.37
SB					
min/week *	11	598.4 (492.7–673.3)	587.8 (498–922.6)	0.929	0.01
min/weekday *	11	650.2 (464–682.6)	596.7 (488.8–1054)	0.929	0.01
min/weekend day *	9	520.5 (380–668.5)	498 (265–552)	0.594	0.13

^†^*p* < 0.05. Note: TEE = total energy expenditure; BMR = basal metabolic rate; MVPA = moderate-to-vigorous physical activity; SB = sedentary behavior; ES = effect size. * Variables expressed as median (minimum and maximum); other variables: mean (±standard deviation).

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
