# Peer review of "Impact of Recreational Sports Activities on Metabolic Syndrome Components in Adolescents"

_ijerph, 2019, doi:10.3390/ijerph17010143_

Round 1
Reviewer 1 Report
The abstract has 2 sentences that should be rearranged in order for clarity. Line 18 "Twelve individuals diagnosed with..." should be placed immediately after the sentence that follows "From this initial sample, 36...." As it is written currently, the reader has to work to figure out that the initial sample is 92, not 12.
I would like to know more about the food (and beverage and snack) consumption, if possible. Perhaps a chart or sample of the items included in the nutrition log. Often kids that age discount sugar-containing beverage consumption, and caffeine (++ sugar) containing beverages. In my experience much of the intake that is adverse happens outside the dining halls, late at night, and on the run. These could also affect some of the variables including weight and HTN directly as well as indirectly.
Author Response
Comment: Review the English language.
Response: One of our co-authors is a native English speaker who has reviewed the paper.
REVIEWER 1
Comment: Change the order of the lines in the Abstract.
Response: Thank you! We found your comments extremely helpful and have revised it.
Comment: To know more about food consumption.
Response: We added more details to this topic.
Reviewer 2 Report
-the scientific standard is high
-general aspects in the "Introduction" could be worked out more clearly
-results are comprehansibly
Author Response
Comment: Review the English language.
Response: One of our co-authors is a native English speaker who has reviewed the paper.
REVIEWER 2
Comment: General aspects of the "Introduction" could be worked out more clearly.
Response: We tried to work out some aspects of the Introduction as suggested.
Reviewer 3 Report
This manuscript aimed to examine the impact of recreational sports program on metabolic syndrome components in Brazilian adolescents. My comments/questions are as follows: 1. In the Abstract part, the body mass index (BMI) was calculated from weight and height (the kg/m2 ratio), not were evaluated BMI. Recreational sports activities contributed to increasing the time spent in MVPA, this is common sense, but not your research conclusion. 2. In the introduction part, this part should be improved. SB is different from PA, you should mention more about it. 3. In the method part, you should add SB measurement. 4. In the statistical analysis part, I suggest that you add regression model Covariates. 5. In the result part, your tables make me a little confusion. Why you used SD in some of variable and use range? In the table 3, why BMI Coef. has two times? In the table 4, pre and post I suggest that age, height, weight and BMI should be deleted it, also height pre 1.73(±0.07) and post 1.76(±0.07)? Does make sense? Last, too many tables, I suggest you merge some tables. 6. In the conclusion part, I suggest that delete some specific results in the conclusion.Author Response
Comment: Review the English language.
Response: One of our co-authors is a native English speaker who has reviewed the paper.
REVIEWER 3
Comment: Recreational sports activities contributed to increasing the time spent in MVPA, this is common sense, but not your research conclusion.
Response: Thank you so much for the suggestion. We agreed and included it in the conclusion.
Comment: Sedentary behavior is different from physical activity, you should mention more about it in the Introduction.
Response: Thank you for this observation. We included it in the introduction.
Comment: In the method part, you should add sedentary behavior measurement.
Response: The sedentary behavior is already in the methodology.
Comment: Why you used SD in some of the variable and use range?
Response: We added the SD to show the range of the data because we judged it would be important for the paper.
Comment: In table 3, why BMI Coef. has two times?
Response: There are two coefficients of BMI because one is showed as anthropometric indicators and the other as obesity variables.
Comment: In the table 4, pre and post I suggest that age, height, weight and BMI should be deleted it, also height pre 1.73(±0.07) and post 1.76(±0.07)?
Response: Thank you. We removed them from the Table.
Comment: In the conclusion part, I suggest that delete some specific results in the conclusion.
Response: Thank you. We removed it from the conclusion.
Round 2
Reviewer 3 Report
The authors did a good job in responding to my comments. However, there are a few concerns that remain to be addressed.
Your mainly topic is to estimate the relationship between recreational sports activities and metabolic syndrome components. In the first review round, I have suggested that deleted the conclusion recreational sports activities contributed to increasing the time spent in MVPA. This is common sense and make no sense to state this point in the conclusion in Line 28. I suggest you could revise it. I suggest that you should use SD not range in the table 1-2, table 5 in the first round. Again, I hope you can revise it.Author Response
Dear Reviewer,
We are grateful for your time and constructive comments on our manuscript. We have implemented their comments and suggestions and wish to submit a revised version of the manuscript for further consideration in the journal.
Changes in the initial version of the manuscript are highlighted in the revised version. Below, we also provide a point-by-point response explaining how we have addressed each of your comments.
We hope that you find our responses satisfactory and that the manuscript is now acceptable for publication.
Yours sincerely,
On behalf of the co-authors.
Comment 1: Remove the line about the impact of the recreational activities on the moderate-to-vigorous physical activity in the conclusion.
Response: thank you for the suggestion. We changed this topic on the manuscript.
Comment 2: Change the Tables 1, 2 and 5, exchanging the value range with SD.
Response: Part of the data on these tables are shown by SD and part by range. We have chosen to display the data like that because of the normality of the values. The data with normal distribution are shown with SD, while the non normal data are presented with the value range. We did that because it's a statistical rule, depending on type of data (normal or non normal).
